



# Atmospheric Chemistry, Sources, and Sinks of Carbon Suboxide, $C_3O_2$

Stephan Keßel,[1] David Cabrera-Perez,[1] Abraham Horowitz,[1] Patrick R, Veres,[1,2] Rolf Sander,[1] Domenico Taraborrelli,[1,3] Maria Tucceri,[1] John Crowley,[1] Andrea Pozzer,[1] Luc Vereecken,[1,3] Jos Lelieveld,[1] Jonathan Williams[1]

[1]Max Planck Institute for Chemistry, Hahn-Meitner-Weg 1, Mainz, Germany
[2]now at: NOAA ESRL Chemical Sciences Division, Boulder, USA and Cooperative Institute for Research in Environmental Sciences, University of Colorado, Boulder, CO, USA
[3]now at: Institute of Energy and Climate Research, IEK-8: Troposphere, Forschungszentrum Jülich, Germany

*Correspondence to*: J. Williams (J.Williams@mpic.de)

**Abstract.** Carbon suboxide, O=C=C=C=O, has been detected in ambient air samples and has the potential to be a noxious pollutant and oxidant precursor; however, its lifetime and fate in the atmosphere is largely unknown. In this work, we collect an extensive set of studies on the atmospheric chemistry of $C_3O_2$. Rate coefficients for the reactions of $C_3O_2$ with OH radicals and ozone were determined using relative rate techniques as $k_4 = (2.6 \pm 0.5) \times 10^{-12}$ cm$^3$ molecule$^{-1}$ s$^{-1}$ at 295 K (independent of pressure between ~25 and 1000 mbar) and $k_6 < 1.5 \times 10^{-21}$ cm$^3$ molecule$^{-1}$ s$^{-1}$ at 295 K. A theoretical study on the mechanisms of these reactions indicates that the sole products are CO and $CO_2$, as observed experimentally. The UV absorption spectrum and the interaction of $C_3O_2$ with water were also investigated, enabling photodissociation and hydrolysis rates to be assessed. The role of $C_3O_2$ in the atmosphere was examined using in-situ measurements, an analysis of the atmospheric sources and sinks, and simulation with the EMAC atmospheric chemistry – general circulation model. The results indicate sub-pptv levels at the Earth's surface, up to about 10 pptv in regions with relatively strong sources, e.g. by biomass burning, and a mean lifetime of ~3.2 days. These predictions carry considerable uncertainty, as more measurement data are needed to determine ambient concentrations and constrain the source strengths.

## 1 Introduction

Carbon suboxide was synthesized for the first time in 1906 by Diels and Wolf (1906), though it is thought to have played a role in experiments at the end of the 19th century by Brodie (1872). It is reported as a poisonous gas at room temperature (boiling point 279.9 K) with a noxious smell that can irritate eyes, nose and airways (Reyerson and Kobe, 1930). The structure of $C_3O_2$ (IUPAC: propadiene-1,3-dione) is a quasi-linear cumulene, O=C=C=C=O, with a shallow W-shape which readily deforms by bending vibrations, and a weak dipole moment (Karyakin et al., 1982; Winnewisser et al., 2006). Experimental and theoretical work has considered the photochemical properties of this molecule (Bayes, 1961; Smith et al., 1966; Masiello et al., 2005; Vander Auwera et al., 1991), as well as its thermodynamic and chemical properties (Winnewisser et al., 2006;





McDougall and Kilpatrick, 1965; Koput, 2000; Ramasami, 2007; Kolbanovskii et al., 2014), and carbon suboxide has found some uses in organic synthesis (Kappe and Ziegler, 1974) and polymers (Carofiglio et al., 1986). It has also been shown to occur as an intermediate in abiotic degradation of aromatics in soils (Huber et al., 2007), is emitted by incomplete combustion of biofuels and biomass (Hucknall, 1985; Roblee et al., 1961), and could occur in extra-terrestrial environments (Bennett et

al., 2008; Huntress et al., 1991; Oyama and Berdahl, 1979).

Due to its hydrolysis to malonic acid (Diels and Meyerheim, 1907; Diels and Wolf, 1906), $C_3O_2$ was not expected to be present in the atmosphere at measurable concentrations, and its relevant chemistry has therefore not been studied extensively. The only pertinent study we are aware of is by Faubel et al. (1977), who measured the rate coefficient of $C_3O_2$ with OH radicals between 295 and 480K, obtaining a rate coefficient of $k(T) = (1.1\pm0.5)\times10^{-11} \times \exp(-(620\pm160)K/T)$ cm$^3$ molecule$^{-1}$ s$^{-1}$.

In this work, we investigate the atmospheric chemistry of carbon suboxide. Its spectroscopic properties are studied at infra-red and ultraviolet wavelengths, to assess its greenhouse potential and to obtain the photolysis rates in the atmosphere. The reaction with OH radicals and, for the first time, $O_3$, is examined, quantifying the rate coefficients, and identifying the products in a theory-based analysis. We also present the first determination of its Henry's law constant, $K_H$, and its hydrolysis rate coefficient, $k_{hyd}$. Additionally, in-situ mass-spectrometric studies show the presence of $C_3O_2$ in the atmosphere. Finally, these

characteristics are incorporated in a global chemistry - general circulation model, examining the $C_3O_2$ budget in the atmosphere.

## 2 Methodologies

### 2.1 Chemicals

$C_3O_2$ was prepared by the low temperature dehydration of malonic acid by $P_4O_{10}$ in a similar manner to described previously

(Diels and Meyerheim, 1907; Long et al., 1954), and diluted with nitrogen bath gas (Westfalen, $N_2$ 5.0) to atmospheric pressure; yielding 20.0 to 26.7 mbar $C_3O_2$ in 1000 mbar $N_2$. No efforts were undertaken to increase the yield of $C_3O_2$ by careful drying of the reactants as performed previously (Diels and Lalin, 1908), as only small concentrations were needed. This method of synthesis also yields $CO_2$ as we show later. The dilute carbon suboxide mixture thus obtained proved to be stable at room temperature, and no polymerisation products were observed even after months of storage.

$O_3$ was made by electrical discharge through oxygen (Westfalen, 5.0) and trapped and stored at dry-ice temperature on silica-gel. $CH_3ONO$ was made by the dropwise addition of 50 % $H_2SO_4$ to a saturated solution of $NaNO_2$ and $CH_3OH$, vacuum distilled and stored at -40 °C.

### 2.2 Henry's law constant and hydrolysis rate

A 50 cm$^3$ min$^{-1}$ flow of synthetic air was directed over the malonic acid + phosphorus pentoxide synthesis mixture (see above),

mixed with different flows of moistened air (20, 50, 70, 100, 150, 200 and 250 cm$^3$ min$^{-1}$ (STP)) to create a flow with different





$C_3O_2$ concentrations, and finally bubbled through a 10 mL buffer solution (Alfa Aesar) with pH values (±0.01) of 2.00, 4.00, 6.00, and 8.00. The bubbler efficiently mixes the airstream with the buffer solution, allowing the carbon suboxide to interact fully with the solution. The remaining airflow was routed through the PTR-TOF-MS (see below) to measure the resulting loss of $C_3O_2$.

### 2.3 Reaction of $C_3O_2$ with OH and $O_3$

The rate constant for the reaction of OH with $C_3O_2$ was investigated by the relative-rate method in an apparatus that has previously been described in detail (Crowley et al., 1999; Raber and Moortgat, 2000). The reaction took place in a 44 L cylindrical quartz reaction chamber equipped with fluorescent lamps (Phillips, TL12) to provide UV radiation (~270–380 nm) to initiate OH formation. The air around the lamps was ventilated in order to avoid temperature increases in the reactor during photolysis. Gas-phase concentration determination of $C_3O_2$ and the reference reactant (ethene, see below) was by infrared absorption spectroscopy using a Bomem DA 008 FTIR spectrometer with an MCT detector. The IR analysis light passed through the reactor 24 times to give a total optical path-length of ~29 m. Absorption spectra were recorded at a resolution (Hamm apodized) of 0.5 cm$^{-1}$. Generally, 32 scans (254 for reference spectra) were co-added to achieve a good signal-to-noise ratio. The infra-red (IR) absorption spectrum (at 0.5 cm$^{-1}$ resolution) of $C_3O_2$ is available in the supplementary information. The rate constant for reaction of $C_3O_2$ with $O_3$ was measured in the same apparatus. In this case, the rate of decay of $C_3O_2$ in a large excess of $O_3$ was monitored by IR absorption as was the formation of reaction products.

### 2.4 UV Absorption spectrum of $C_3O_2$

The UV absorption spectrum of $C_3O_2$ was measured in an optical set up that has been described previously (Dillon et al., 2006). Briefly, flowing samples of $C_3O_2$ passed through a 110 cm long optical absorption cell equipped with external, multi-pass white optics aligned to give a total absorption path length of 880 cm. Analysis light from a $D_2$ lamp was collimated and transmitted (through quartz windows) along the axis of the cell before being detected by a 0.5 m monochromator / diode array spectrometer with an effective resolution of 1 nm. Pressures in the cell were monitored with a 13.3 mbar (10 Torr) capacitance manometer. Wavelength calibration and measurement of the spectral resolution were achieved using lines from a low-pressure Hg lamp.

### 2.5 PTR-TOF-MS measurements

The PTR-TOF-MS (Proton Transfer Reaction Time of Flight Mass Spectrometer, Ionicon Analytik GmbH, Innsbruck, Austria) has been described in detail elsewhere (Graus et al., 2010). Briefly, this measurement technique is based on the protonation of molecules by $H_3O^+$ ions that are generated in a hollow cathode discharge. Within the time of flight mass spectrometer, the reagent ions are accelerated through ambient air and the resulting molecular ions brought to an equivalent kinetic energy level, such that their subsequent velocity in the flight tube depends on the mass-to-charge ratio (m/z). By measuring the "time-of-flight", the mass-to-charge ratio can be calculated at high mass resolution. The mass resolution was approximately 3700 $m/\Delta m$.





The instrument was operated with a drift pressure of 2.20 hPa (E/N 140 Td) and a drift voltage of 600 V. 1,3,5-trichlorobenzene was used as internal standard for mass calibration. Data post-processing and analysis was performed by using the program "PTR-TOF data analyzer", which is described elsewhere (Müller et al., 2013). Since the measured mixing ratios of $C_3O_2$ were low it was necessary to integrate the signal over extended periods of time (e.g. 1 hour).

The mass resolution of this instrument is essential to separate the protonated carbon suboxide signal ($m/z = 68.998$) from other compound signals with the same nominal mass, such as furan ($m/z = 69.034$) and isoprene ($m/z = 69.070$). To ensure these compounds could be separated, a synthetic mixture of furan (Acros Organics, 99%), isoprene (Air Liquide), and carbon suboxide in synthetic air (Westfalen, 20.5% $O_2$, 79.5% $N_2$) was analyzed (see Figure 1), showing clear peak separation. In-situ sampling was made through Teflon tubing. The inertness of the Teflon surfaces was verified by varying the tubing length and

observing no change in the carbon suboxide signal.

### 2.6 Theoretical calculations

To further explore the reactions of carbon suboxide, we employed quantum chemical calculations, mainly using the M05-2X DFT and M06-2X density functionals (Zhao et al., 2006; Zhao and Truhlar, 2008) with the aug-cc-pVTZ basis set (Dunning, 1989), supplemented by CBS-QB3 (Montgomery et al., 1999) or ROHF-UCCSD(T)/aug-cc-pVTZ single point energy

calculations for a subset of the intermediates. The DFT and CBS calculations were performed with Gaussian-09 (Frisch et al., 2010), the ROHF-CCSD(T) calculation with Molpro-2010.1 (Werner et al., 2010). These levels of theory are expected to be accurate within ~2 kcal mol$^{-1}$ for relative energies; this is sufficient to identify the main reaction channels in the atmospheric oxidation of $C_3O_2$. If smaller uncertainty intervals are needed in a future study, the geometries listed in the supporting information are expected to be of sufficient quality for more accurate single-point energy calculations. Rate coefficient

predictions were performed using Transition State Theory in a harmonic oscillator rigid rotor approximation (Pilling and Seakins, 2007; Truhlar et al., 1996; Vereecken et al., 2015), and exploratory master equation calculations were performed using the URESAM software (Vereecken et al., 1997).

### 2.7 Global model simulations

In this work, the ECHAM/MESSy Atmospheric Chemistry (EMAC) model was used to numerically simulate the $C_3O_2$

distribution in the atmosphere, and to estimate its budget. The EMAC model is a numerical chemistry and climate simulation system that includes sub-models describing tropospheric and middle atmosphere processes and their interaction with oceans, land and human influences (Jöckel et al., 2010). It uses the second version of the Modular Earth Submodel System (MESSy2 version 2.42) to link multi-institutional computer codes. The core atmospheric model is the 5th generation European Centre Hamburg general circulation model (ECHAM5 version 5.3.02) (Roeckner et al., 2006). For the present study we applied

EMAC at the T42L31 resolution, i.e. with a spherical truncation of T42 (corresponding to a quadratic Gaussian grid of approx. 2.8 by 2.8 degrees in latitude and longitude) with 31 vertical hybrid pressure levels up to 10 hPa. In this study the same set-up evaluated in previous studies was used (Yoon and Pozzer, 2014; Pozzer et al., 2015), and only the following submodels were



modified for the simulation of $C_3O_2$ : MECCA (Sander et al., 2011), JVAL (Sander et al., 2014), SCAV (Tost et al., 2006), DRYDEP (Kerkweg et al., 2006a, 2009), and OFFEMIS (Kerkweg et al., 2006b). The numerical simulation performed covers the years 2005-2006, with the first year used as spin-up. The results presented here are representative for the meteorology and emissions (e.g. biomass burning) for the year 2006. The prescribed biomass burning emissions used were those based on The

Global Fire Assimilation System (GFASv3.0) (Kaiser et al., 2012), with daily resolution and $0.5° \times 0.5°$ spatial resolution. The emissions factors (in g/kg) for carbon suboxide are those from literature (Yokelson et al., 2013) where available. The emission factor provided by Yokelson et al. for semiarid shrublands ($1.2\times10^{-3}$ g/kg), coniferous canopy ($1.2\times10^{-3}$ g/kg) and organic soil ($3.75\times10^{-3}$ g/kg) are assumed to also be representative for savanna, extra-tropical forest and peat, respectively. For agricultural and tropical forest emission factors, no values are available, and we selected an emission factor of $1.0\times10^{-3}$ g/kg,

of similar magnitude as the other factors. Further emissions from anthropogenic biofuel consumption were added using the EDGARv4.3 database (Crippa et al., 2016). The amount of consumed biofuel was obtained by dividing the emissions of non-methane volatile organic hydrocarbons (NMVOC) from biofuel usage by the emission factor for NMVOC as listed in Yokelson et al. (2013), i.e. 26.44 g/kg. The emissions of $C_3O_2$ from this consumption were then again estimated using the previously mentioned factor ($1.0\times10^{-3}$ g/kg). This could be a low estimate as biofuel combustion may resemble the burning of peat more

than of vegetation, but this will need to be tested experimentally.

## 3 Results

### 3.1 UV Absorption spectrum of $C_3O_2$

The UV absorption (230 to 309 nm) at various pressures of $C_3O_2$ was measured in the optical absorption cell (892 cm path length) described above. According to the Beer-Lambert law, for a given optical path length ($l$, cm) the optical density ($OD$)

of an absorbing sample is proportional to its concentration [C] (in molecule cm$^{-3}$):

$$OD = \ln (I_0/I) = \sigma(\lambda)l[C] \qquad (1)$$

where $\sigma(\lambda)$ is the wavelength dependent absorption cross section (cm$^2$ molecule$^{-1}$). Thus, knowledge of the absolute concentration of $C_3O_2$ was necessary to derive its absorption spectrum. Our experiments using the FTIR apparatus revealed (see below) that substantial levels (up to 80 %) of $CO_2$ impurity are present in the $C_3O_2$ samples prepared as described above.

Mass spectrometer analysis (electron impact, 70 eV) revealed no further impurities.

Whilst $CO_2$ impurities do not represent a problem for the relative rate constant measurements described below, it does mean that sample pressure cannot be directly converted to a concentration. For this reason, each sample for which the UV absorption was to be measured was first analysed for its $CO_2$ impurity using the FTIR apparatus described earlier. For a given pressure (P) of the $C_3O_2$ / $CO_2$ sample introduced into the UV absorption apparatus, the fractional contribution of $C_3O_2$ was calculated

and then converted to a concentration [C] using Boyles law ([C] = $P N_A/RT$). As $CO_2$ does not absorb light between 230 and 309 nm, it does not contribute to the overall optical density. The dependence of the measured optical density at 264.8 nm on





the $C_3O_2$ concentration is plotted in Figure 2. The expected Beer-Lambert linearity is observed, the slope of the plot, $\sigma(264.8$ nm)$\cdot l$, yielding an absorption cross section at this wavelength of $\sigma(264.8$ nm$) = 3.08 \times 10^{-19}$ cm$^2$ molecule$^{-1}$. The total uncertainty for these experiments is related to the uncertainty associated with the correction applied for the large $CO_2$ impurity, which was $80 \pm 5$ %. The contribution of $C_3O_2$ was thus $20 \pm 5$ %, i.e. with an error of 25 %. We therefore quote a final value

of $\sigma(264.8$ nm$) = 3.1 \pm 0.8$ cm$^2$ molecule$^{-1}$. In order to accurately measure optical absorption on the long wavelength wing of the $C_3O_2$ spectrum (i.e. at $\lambda > 310$ nm, where tropospheric, actinic radiation starts to become important) absorption spectra were measured using large amounts of $C_3O_2$. The absorption spectrum (black line) presented in Figure 3 is thus a composite of three different, overlapping measurements of optical density (230-293, 293-299 and 299-330 nm), normalized to the cross section derived at 264.8 nm; the underlying data is available in the supporting information.

Figure 3 also plots the previously reported spectrum of $C_3O_2$, where Bayes (1962) reports a value of $\sigma_{max} = 3.59 \times 10^{-19}$ cm$^2$ molecule$^{-1}$ at 265 nm, which is ~ 17 % larger than the value obtained in this work. Indeed, at all wavelengths longer than 250 nm, a scaling factor of 1.17 brings our data into very good agreement with the spectrum reported by Bayes. At shorter wavelengths, especially less than 245 nm, there is substantial disagreement and our cross sections are larger. The difference may be explained by additional, absorbing impurities in our sample, or are due to working with low light intensities resulting

from the multiple-reflection optical set-up. However, these wavelengths are not relevant for the tropospheric chemistry of $C_3O_2$. Bayes performed multiple distillations of his sample, which was therefore presumably purer. We therefore choose to scale our spectrum by a factor 1.17 which results in the thin black line in Figure 3 and extends the wavelength range out to 330 nm.

## 3.2 The reaction of $C_3O_2$ with OH radicals

### 3.2.1 Experimental relative rate study

OH radicals were generated by photolysing $CH_3ONO$ (270-380 nm) in air in the presence of NO:

$$CH_3ONO + h\nu \quad \rightarrow CH_3O + NO \qquad\qquad (R1)$$
$$CH_3O + O_2 \qquad \rightarrow HCHO + HO_2 \qquad\quad (R2)$$
$$HO_2 + NO \qquad \rightarrow OH + NO_2 \qquad\quad (R3)$$

OH thus formed reacted with $C_3O_2$ and $C_2H_4$, both of which were present in the initial reaction mixture:

$$OH + C_3O_2 \qquad \rightarrow products \qquad\qquad (R4)$$
$$OH + C_2H_4 \qquad \rightarrow products \qquad\qquad (R5)$$

Typical concentrations of $C_3O_2$ and $C_2H_4$ were 2-3 $\times 10^{13}$ molecule cm$^{-3}$ and 5-12 $\times 10^{14}$ molecule cm$^{-3}$, respectively. We note that, for the kinetic analysis described below, knowledge of absolute concentrations is not necessary. Rather, the relative rate

technique relies only on accurate measurement of depletion factors for both $C_3O_2$ and $C_2H_4$. When adopted for the present reactions, the expression for deriving the relative rate constant ($k_4 / k_5$) is (Pilling and Seakins, 2007) :





$$\ln\frac{\left[C_3O_2\right]_{t=0}}{\left[C_3O_2\right]_t} = \frac{k_4}{k_5}\cdot\ln\frac{\left[C_2H_4\right]_{t=0}}{\left[C_2H_4\right]_t} \qquad (2)$$

where $k_4$ and $k_5$ are the bimolecular rate constants for reaction of OH with $C_3O_2$ (R4) and $C_2H_4$ (R5) respectively, at the selected temperature and pressure. $[X]_0$ and $[X]_t$ represent concentrations of the non-radical reactants X at time zero (i.e. before OH is generated) and reaction time $t$, respectively. Figure 4 shows the change in absorption due to $C_3O_2$ and $C_2H_4$ following several

photolysis periods, intermediate to which the FTIR spectra were obtained. $C_3O_2$ was monitored via its strongest vibrational feature, the asymmetric stretch ($\upsilon_3$) at ~2260 cm$^{-1}$ (Miller and Fateley, 1964). The reaction was allowed to proceed until $C_2H_4$ was depleted to $\leq$ 45 % of its original concentration, whereby $C_3O_2$ was depleted to $\leq$ 77 % of its original concentration. In Figure 4, the depletion of $C_3O_2$ and $C_2H_4$ is seen to be accompanied by formation of both CO (fine rotational structure around 2200 cm$^{-1}$) and $CO_2$ (fine rotational structure around 2300 cm$^{-1}$). Typically, the photolysis lamps were powered for 2 minute

periods, the FTIR spectra (32 scans at 0.5 cm$^{-1}$ resolution) took about 1.5 minutes. Without illumination by the TL12 lamps, no depletion of $C_3O_2$ or $C_2H_4$ could be observed so that significant reaction with e.g. surfaces or $CH_3ONO$ or NO or radicals formed in the dark could be ruled out. In addition, illumination of a gas-mixture containing $C_3O_2$ and $C_2H_4$, without $CH_3ONO$, did not lead to observable depletion of the reactants. Accurate analysis of relative depletion factors requires that the infra-red absorption features used are linear with concentration and that no products absorb significantly at the same wavelength. Figure

5 shows the results of an experiment in which both $C_3O_2$ and $C_2H_4$ were depleted by reaction with OH. The depletion factor after 8 minutes of reaction was then derived by least squares fitting of the $C_3O_2$ and $C_2H_4$ spectra obtained after reaction to the spectra obtained before. As indicated by the lack of structure in the residuals, there is no evidence for infra-red active products in the narrow spectral regions used to analyse the data.

The $C_3O_2$ and $C_2H_4$ depletion factors are plotted against each other in Figure 6 for experiments carried out at 295 $\pm$ 2 K and

pressures of 25, 399 and 1003 mbar. The excellent linearity confirms that the experimental procedure is appropriate, the slopes of the fit lines providing the relative rate constant ($k_4$ / $k_5$) at each pressure, which are listed in Table 1. The rate constant $k_4$ obtained by using the recommended value of $k_5$ (IUPAC Subcommittee on Atmospheric Chemical Kinetic Data Evaluation, 2015) displays no significant dependence on pressure. The slightly lower value at 25.3 mbar may be the result of a weak non-linearity in the absorbance of $C_2H_4$ at low pressure where the vibrational features narrow in comparison to the instrumental

resolution. This was not observed at the higher pressures where pressure broadening increases the line widths. For this reason, we prefer to quote a final, pressure independent, rate constant at 295 K of $(2.6 \pm 0.5) \times 10^{-12}$ cm$^3$ molecule$^{-1}$ s$^{-1}$.

This value can be compared to the single previous determination of $k_4$ that we are aware of (Faubel et al., 1977), in which $k_4 = (1.2 \pm 0.5) \times 10^{-12}$ exp(-620K/$T$) cm$^3$ molecule$^{-1}$ s$^{-1}$ was reported, resulting in a value of $(1.4 \pm 0.6) \times 10^{-12}$ cm$^3$ molecule$^{-1}$ s$^{-1}$ at 295 K. Even within the combined uncertainty, this result disagrees with our larger value of $2.6 \pm 0.5 \times 10^{-12}$ cm$^3$ molecule$^{-1}$

s$^{-1}$. Faubel et al. (1977) used an absolute method (detection of OH using ESR) to measure the rate constant at low pressure (2.7-4 mbar He). They did not vary the reaction time (fixed at 16 ms) in their experiments, but monitored the change in OH concentration as various amounts of $C_3O_2$ were added. Errors in this measurement will be associated with the fact that wall



losses of OH could not be accurately assessed by varying the contact time, and that by using high concentrations of OH, the self-reaction (to form O and H atoms) could also take place to a significant extent. They also observed that the depletion of $C_3O_2$ (measured using mass spectrometry) relative to OH was only a factor of 0.5 and used this factor to scale the rate constant derived from observation of OH loss only, which was $2.8 \times 10^{-12}$ cm$^3$ molecule$^{-1}$ s$^{-1}$ at 295 K. Without this scaling factor, the

results would be in excellent agreement. While the factors above may play a role, it is difficult to rigorously assess the causes of the disagreement. We note, however, that the relative-rate method has significant advantages over absolute methods when reactant concentrations are difficult to define accurately. In the case of Faubel et al. (1977) this is especially true as both OH and $C_3O_2$ concentrations needed to be known. For the purposes of atmospheric modelling of the reaction between OH and $C_3O_2$, the rate constant obtained using the relative rate technique (at atmospheric pressure) is preferred.

The only (IR-active) products observed in the present study of OH + $C_3O_2$ were CO and $CO_2$. As both CO (directly) and $CO_2$ (indirectly) are also products of the photolysis of HCHO (formed in reaction R2) we did not attempt an analysis of the product yields. We note that the formation of only CO and $CO_2$ as products is consistent with the observations of Faubel et al. (1977) and with our theoretical analysis of this reaction (see below).

### 3.2.2 Theoretical study of the reaction mechanism

Two distinct addition sites exist for the OH radical on carbon suboxide (see Figure 7). Addition on the central carbon, forming an acyl radical, has the lowest entrance barrier of ~1 kcal mol$^{-1}$, though it leads to the least stable adduct **INT1** with an exoergicity of only 17.8 kcal mol$^{-1}$, compared to the 31.1 kcal mol$^{-1}$ potential energy release for addition on the outer carbons to **INT7** with an entrance barrier of ~ 4 kcal mol$^{-1}$. The underlying reason is that **INT7** is stabilized mostly by resonance, which only becomes active after the TS is traversed and the radical electron is freed to delocalize. Both reaction channels pass through

a shallow pre-reactive complex, approximately 1.5 kcal mol$^{-1}$ below the reactants. Based on ROHF-CCSD(T)//M06-2X/aug-cc-pVTZ data, we obtain a high-pressure rate coefficient of $5.6 \times 10^{-12}$ cm$^3$ molecule$^{-1}$ s$^{-1}$ at 295 K, in agreement with the experimental value within the expected uncertainty on the theoretical result. Here, the reaction proceeds nearly exclusively (>99% at 295K) by addition to the central carbon. We have attempted to investigate the pressure-dependence of this reaction using exploratory Master Equation analysis. Using a well depth equal to the stability of **INT1**, we obtain a pressure dependence

that far exceeds the experimental data. Using an energy well depth for addiction equal to **INT7**, on the other hand, leads to results that are fully compatible with the experimental data; in this scenario, the differences in rate coefficient of Faubel et al. (1977) at 2.7 mbar, and the current results at pressures between 25.3 and 1003 mbar is due solely to fall-off. Unfortunately, this latter scenario does not match our current understanding of the reaction mechanism, and we are unable to reconcile the theoretical mechanism against the experimental fall-off without computational cost-limiting additional calculations. As such,

we do not make a recommendation for the pressure-dependence of this reaction, and state only that the reaction is near the high-pressure limit at atmospheric pressures.

In the atmosphere, the acyl radical **INT1**, formed from addition of OH on the central carbon, readily adds an $O_2$ molecule to form acetylperoxy radical **INT2** (see Figure 7), which is expected to react similar to other acetylperoxy radicals. In a NOx-




rich environment, this peroxy radical is oxidized to an acyloxy radical **INT3**, which loses a $CO_2$ molecule to form the α-OH-vinoxy radical **INT4**. The dominant resonance structure of this radical is the alkyl radical, which can undergo an addition-elimination reaction with $O_2$ to form an $HO_2$ molecule and **INT5**. This latter product is unstable on the singlet PES, decomposing to 2 CO molecules. On the triplet surface **INT5** is very short-lived, and will undergo an intersystem crossing to

the singlet surface where it decomposes. A fraction of **INT4** could undergo an $O_2$ addition to **INT6**, followed by reaction with NO to form acyl radical **INT12**, which oxidizes further to 2 $CO_2$ and $HO_2$.

The addition on the outer carbon forms a carboxylic acid **INT7**, with an ethynoxy-resonance stabilized radical site (see Figure 7). Decomposition to $C_2O$ + HOCO or H-migration in **INT7** have high barriers of 90 and 42 kcal mol$^{-1}$, respectively, leaving $O_2$ addition as the dominant atmospheric reaction channel, with an exoergicity of 25.1 kcal mol$^{-1}$. The resulting peroxy radical

**INT8**, OCC(OO•)COOH, has a fairly mobile hydrogen atom, which can temporarily migrate to the oxygen radical site; this H-shift is endothermic by 11 to 12 kcal mol$^{-1}$. The resulting hydroperoxide acyloxy radical, OCC(OOH)C(O•)O, is not an energetic minimum at the chosen level of theory, spontaneously reforming the carboxylic acid peroxy radical; this is contrary to similar reactions in aliphatic carboxylic acid atmospheric oxidation where $CO_2$ is preferentially eliminated (da Silva, 2010). All reactions of the transient hydroperoxide examined, including $CO_2$ elimination and cyclisation, appear to have relatively

high barriers exceeding 20 kcal mol$^{-1}$ relative to the peroxy radical. We propose that the **INT8** alkylperoxy radical predominantly undergoes the traditional atmospheric $RO_2$ reaction, i.e. bimolecular reactions with NO, $HO_2$ and $RO_2$; we will focus here on a more polluted environment, where reaction with NO is the dominant fate, forming an oxy radical **INT9**. Unimolecular reaction in this oxy radical, i.e. HOCO or CO elimination, and H-migration, all show barriers in excess of 17 kcal mol$^{-1}$, again leaving $O_2$ addition as the dominant reaction channel, forming the **INT10** acetylperoxy radical. This radical

will undergo the reaction traditional for acetylperoxy radicals in the atmospheric, i.e. in a NOx-rich environment **INT10** will be oxidized to 3 $CO_2$ + $HO_2$. When considering competing reactions of the peroxy radical with $RO_2$ and $HO_2$, highly oxidized compounds bearing multiple carbonyl and carboxylic acid functionalities can be formed instead.

We conclude that the OH-initiated oxidation of $C_3O_2$ leads mainly to CO and $CO_2$, with regeneration of a $HO_2$ radical; depending on the reaction conditions, some OVOC can be formed in cross-reactions of $RO_2$ intermediates with $HO_2$ or $RO_2$.

The chemistry of some intermediates in the current oxidation scheme receives further attention in the supporting information.

### 3.3 The reaction of $C_3O_2$ with $O_3$

#### 3.3.1 Experimental study of the rate coefficient

The rate constant for reaction of $C_3O_2$ with $O_3$ ($k_6$) was measured in the same apparatus as used for the OH + $C_3O_2$ study.

$$O_3 + C_3O_2 \quad \rightarrow products \qquad (R6)$$

In this case, a large concentration of $O_3$ was eluted into the reactor from its storage vessel ($O_3$ was stored on silica gel at 196 K, see above) and its concentration monitored using calibrated absorption features (2001-2112 cm$^{-1}$, 901-995 cm$^{-1}$ and 702-798 cm$^{-1}$). $O_3$ concentrations were varied between 0.1 and 2.1 mbar. $C_3O_2$ was subsequently added and the mixture brought





up to atmospheric pressure with synthetic air. The mixture was then allowed to stand in the dark, the relative $C_3O_2$ concentration change being monitored at regular intervals by FTIR. The rate constant is low, so long reaction times and high concentrations of $O_3$ were used.

Figure 8 displays the results of an experiment in which ~ $2.7 \times 10^{-3}$ mbar $C_3O_2$ was mixed with 1 mbar of $O_3$. The spectra shown

were referenced to the spectrum obtained directly after $C_3O_2$ was added to the mixture so that, if reaction takes place, reactants should display increasingly negative absorptions and products increasingly positive absorptions as time progresses. It is readily seen that, following reaction times exceeding ~30 mins, significant depletion of $C_3O_2$ was observed (negative absorption feature centered at 2260 cm$^{-1}$) and at the same time CO and $CO_2$ were formed (positive absorption features centered at ~2140 and 2350 cm$^{-1}$, respectively). No changes were observed at the $O_3$ absorption features owing to the fact that, as $O_3$ was present

in large excess over $C_3O_2$, its concentration did not change significantly over the course of the reaction. In the absence of $O_3$, $C_3O_2$ concentrations did not change significantly over a period of several hours.

Given that $O_3$ is in large excess, and that $C_3O_2$ does not react in the absence of $O_3$, a simple expression can be used to analyse the data:

$$\ln \frac{[C_3O_2]_0}{[C_3O_2]_t} = k_6[O_3] \cdot t \tag{3}$$

Where $k_6$ is the rate constant for reaction R6, $t$ is the reaction time, $[O_3]$ is the ozone concentration and $[C_3O_2]_0 / [C_3O_2]_t$ is the time-dependent, relative change in $C_3O_2$ concentration.

In Figure 9 we plot the relative change in $C_3O_2$ concentration versus reaction time for several experiments using different amounts of $O_3$. All results were obtained at $295 \pm 2$ K and at 1 atm total pressure of air. The good linearity of $[C_3O_2]_0 / [C_3O_2]_t$ versus reaction time suggests that a pseudo-first order analysis is appropriate to analyse the data; the slope of each fit line is

then equal to $k_6[O_3]$. By plotting the values of $k_6[O_3]$ thus obtained against $[O_3]$ we derive the bimolecular rate constant, $k_6$ as shown in Figure 10. The value obtained is $k_6 = (1.5 \pm 0.3) \times 10^{-21}$ cm$^3$ molecule$^{-1}$ s$^{-1}$ where the uncertainty contains an assessment of both statistical and systematic errors. As often the case in absolute kinetic studies conducted in pseudo-first-order conditions, the main source of error is the derivation of the concentration of the excess reagent, in this case $O_3$. Based on differences in derived concentrations using the three $O_3$ absorption features listed above, we estimate that the uncertainty

in the $O_3$ concentration is < 12 %.

We also note that the extrapolated fit to the data in Figure 10 (i.e. the $C_3O_2$ loss rate in the absence of $O_3$) is larger than that experimentally observed. This is potentially due to the dark formation of radicals (e.g. OH) that may react with $C_3O_2$. Indeed, evidence for the formation of small amounts of OH when adding $O_3$ to quartz reactors has been observed previously (Finlayson-Pitts and Pitts, 1999). Taking a value of $k_4 = 2.6 \times 10^{-12}$ cm$^3$ molecule$^{-1}$ s$^{-1}$, we can show that a steady-state OH concentration

of $3 \times 10^6$ molecule cm$^{-3}$ would be required to (positively) bias the rate constant by ~ 10 %. Whilst we have no evidence for this in our reactor, we recognise that such effects are difficult to eliminate for very slow reactions and thus consider the value





of $k_6$ obtained as an upper limit. In any case, as we show later, the rate constant is too low for it to represent a significant sink of atmospheric $C_3O_2$.

The only IR-active products observed from the reaction of $C_3O_2$ with $O_3$ were CO and $CO_2$ as seen in Figure 8. Attempts to derive the yields of CO and $CO_2$ experimentally were unsuccessful as variable amounts of both were formed when (in the

absence of $C_3O_2$) $O_3$ and synthetic air were allowed to stand in the reactor for a few hours, presumably resulting from surface reactions of $O_3$ on the quartz / metal surfaces in the reactor. The amount of CO and $CO_2$ formed in this surface reaction were comparable to those observed when $C_3O_2$ was present.

There is no previous experimental data with which to compare our rate constant, but note that the low value of $k_6$ and the observation of only CO and $CO_2$ as stable products are consistent with the theoretical investigation of this reaction (see below).

**3.3.2 Theoretical study of the reaction mechanism**

The reaction of carbon suboxide with $O_3$ proceeds by a cycloaddition of ozone across one of the equivalent C=C bonds (see Figure 11), with a barrier calculated at 11.4 (M05-2X) to 11.9 kcal mol$^{-1}$ (CBS-QB3). The TST rate coefficient predicted for the addition at room temperature is $6.7 \times 10^{-24}$ cm$^3$ molecule$^{-1}$ s$^{-1}$, based on M05-2X data, in fair agreement with the experimental value considering the low level of theory used. The primary ozonide (**POZ**) is formed with an internal excess energy of 62.5

kcal mol$^{-1}$, and has access to two dissociation channels. The first channel, with a barrier of 27.7 kcal mol$^{-1}$ breaks an O−O bond in the five-membered ring, leading to the formation of $CO_2$ and the **OOC=C=O** Criegee intermediate (CI, carbonyl oxide) in either the *syn* or *anti* conformer. The lowest dissociation channel of the **POZ**, with a barrier of 20.2 kcal mol$^{-1}$, breaks the other O−O bond in the POZ; however, rather than forming the expected **OOC=O** Criegee intermediate, the molecule rearranges into a dicarbonyl cyclic peroxide **INT1** (see Figure 11), splitting of the extracyclic CO moiety. A visualisation of

this unusual rearrangement is available in the supporting information. This reaction is exoenergetic by 72.2 kcal mol$^{-1}$; the high internal energy content of **INT1** leads to facile decomposition to 2 $CO_2$ molecules. The chemistry of **OOC=C=O** depends on its conformation; note that CI have a high barrier for *syn*/*anti* isomerisation (Vereecken and Francisco, 2012), such that the conformers need to be considered as separate chemical species. The *syn* conformer readily undergoes a 1,4-ring closure reaction, with a negligible barrier. The resulting four-membered ring is not a true minimum at the DFT level of theory, falling

apart into CO + $CO_2$. The *anti*-CI conformer has a stronger biradical character, and follows the more traditional CI chemistry, first by cyclising to a dioxirane **DIO**, 17.0 kcal mol$^{-1}$ more stable than the CI, followed by ring opening to the singlet bisoxy (**SBO**) biradical, which is found to be unstable and decomposes without barrier to CO + $CO_2$. The chemistry of some related carbon oxides is discussed briefly in the supporting information.

From these calculations, we conclude that the ozonolysis of carbon suboxide leads to 2 $CO_2$ + CO, irrespective of the channel;

the high exothermicity of the reactions combined with low intermediate reaction barriers suggests a chemically activated decomposition process, with no collisional thermalization of the intermediates at ambient pressure.





### 3.4 Determination of the Henry's law and hydration constants

After passing a flow of $C_3O_2$ through water for about two minutes an equilibrium was reached, such that no more $C_3O_2$ was taken up by the solution. The 50 cm$^3$ min$^{-1}$ carbon suboxide flow was then diverted, allowing the solution to degas with a rate that depends on its Henry's law constant $K_H$ and the hydrolysis constant $k_{hyd}$. The time dependent concentration $c_t$ at time $t$

depends on the gas flow $\phi$, the (constant) volume $V$ of the solution, and the temperature (here $T = 296K$) as follows (Roberts, 2005) :

$$\ln\left(\frac{c_o}{c_t}\right) = t \cdot \left(\frac{\phi}{k_H \cdot R \cdot T \cdot V} + k_{hyd}\right) \qquad (4)$$

The slope of the plot of $\ln(c_0/c_t)$ versus time equals a point on a linear function of gas flow $\phi$ with intercept $k_{hyd}$ and slope $(K_H \cdot R \cdot T \cdot V)^{-1}$, such that the series of flows $\phi$ described above allows the determination of both $k_{hyd}$ and $K_H$. This entire procedure

was repeated for multiple pH values of the solution. Changes in the volume $V$ due to evaporation or physical transport after e.g. bubble bursting is considered negligible. Figure 12 shows the PTR-TOF-MS measurements of the carbon suboxide signal in the air flow as a function of time, showing the adsorption and subsequent evacuation of carbon suboxide from the water volume. Figure 12 also shows a more detailed plot of the evacuation of carbon suboxide from the water volume and the $\ln(c_0/c_t)$ values derived from the $C_3O_2$ concentration. Different synthetic air flows, i.e. different amounts of $C_3O_2$, $\phi/V$, yield different

slopes which were then used through Eq. (4) to determine $K_H$ from the slope $\phi(K_H \cdot R \cdot T \cdot V)^{-1}$, and $k_{hyd}$ from the intercept; this procedure was repeated for a set of pHs. Table 2 shows the results of this analysis, as a function of the pH value of the aqueous solution. From these values, it is clear that carbon suboxide absorbs better, and hydrolyses significantly faster, in basic solutions compared to acid solutions.

### 3.5 Photodissociation of $C_3O_2$ in the atmosphere

Photodissociation of carbon suboxide in the atmosphere depends on the wavelength-specific absorption cross section $\sigma(\lambda, T)$ (see above), quantum yield $\Phi(\lambda, T)$ for dissociation, and actinic flux $F_\alpha(\lambda)$:

$$J_{C_3O_2}(T) = \int_\lambda \sigma_{C_3O_2}(\lambda, T) \cdot \Phi_{C_3O_2}(\lambda, T) \cdot F_\alpha(\lambda) \cdot d\lambda \qquad (5)$$

The photolysis rate was determined using the Landgraf and Crutzen (1998) method, using the following approximations: (i) over the photochemically active spectral range from 178.6 to 752.5 nm, only the measured absorption cross sections (listed in

the supporting information) between 230 and 309 nm are used; all other values are set to zero; (ii) based on the lack of structure in the UV absorption spectrum (indicative of excitation at energies beyond the dissociation limit) a quantum yield of 1 is assumed; (iii) an atmosphere without clouds is assumed, where the actinic flux and the relative humidity were averaged over their values at 19 different heights; (iv) an albedo of 0.07 is assumed. In the calculation, the vertical temperature changes were





accounted for. Under these assumptions, for a zenith angle of 50 degrees, photodissociation rate constants ranging from $1.2 \times 10^{-6}$ to $4.0 \times 10^{-6}$ s$^{-1}$ were obtained.

### 3.6 In-situ measurements of $C_3O_2$

PTR-MS-ToF measurements were performed in several locations with a view to identifying potential $C_3O_2$ sources. These included near the MPI-Chemistry building in Mainz, Germany (anthropogenically influenced continental air), in a tropical greenhouse (strong biogenic source), direct measurement from diesel and gasoline car exhaust, and in volcanic gas emissions (geogenic source). Measurements from the green-house (in-situ) and volcanic air samples (metal canisters of pressurized air) showed no significant peak on m/z = 68.995 of protonated $C_3O_2$, Likewise, flue gases from a petrol (gasoline) and a diesel vehicle, analyzed after dilution, showed no clear signal in either case. However, the ambient air measurement at the MPIC building (in-situ) showed a small peak (Figure 1) indicating the presence of carbon suboxide in ambient air. The measurements therefore show that carbon suboxide can be observed with current technology, and separated effectively from other peaks in the mass spectrum. Earlier work by Jordan et al. (2009) also showed a signal at the $C_3O_2$ mass in ambient air, though these authors did not identify the compound as carbon suboxide. Although the $C_3O_2$ signal was not quantified through species-specific calibration, assuming a calibration factor similar to that of isoprene gives a mixing ratio in ambient air close to the instrument detection limit of about 10 pptv when integrated for one hour. At this time, we do not propose atmospheric $C_3O_2$ sources, but adhere to the single source category described in the literature (biomass burning), discussed in more detail below. Future measurements using PTR-MS-ToF technology should with time reveal whether further sources of $C_3O_2$ exist.

### 3.7 Atmospheric model for $C_3O_2$

In the numerical simulation of carbon suboxide global distribution, we assume that its main sources into the atmosphere are biomass burning and biofuel consumption (see Sec. 2.7). For atmospheric carbon suboxide, the three possible chemical removal paths, discussed earlier in this paper, were implemented. The first reaction is with OH, with a rate constant of $2.5 \times 10^{-12}$ cm$^3$ molecule$^{-1}$ s$^{-1}$, the second is photolysis, and the third, minor reaction is with ozone that has a low rate coefficient of $1.5 \times 10^{-21}$ cm$^3$ molecule$^{-1}$ s$^{-1}$. In addition to these chemical transformations, two possible physical sinks, i.e. wet and dry deposition, are included. Wet deposition follows dissolution into cloud droplets that form precipitation and rain out; the Henry's law constant of carbon suboxide is very low, 1.4 M atm$^{-1}$. Subsequent to uptake into the aqueous phase the molecule is hydrolyzed and forms a carboxylic acid ketene with a rate constant of $4 \times 10^{-2}$ s$^{-1}$ (see above). Further hydrolysis yields malonic acid, with a rate constant of 44 s$^{-1}$, as determined for ketene hydrolysis (Bothe et al., 1980). This reaction chain is rate limited by the first hydrolysis step, and the hydrolysis system was implemented as a single lumped process. Overall, since only a small fraction of carbon suboxide partitions into hydrometeors, wet deposition is expected to be of less significance than chemical conversion by OH in the gas phase.

Table 3 quantifies annual fluxes of carbon suboxide emissions and sinks in Gg/year. Potential underestimation of the emissions may be related to small undetected fires. The main sink of carbon suboxide is reaction with OH, which accounts for more than





the 59 % of the loss. Conversion of carbon suboxide to malonic acid in the aqueous phase is the second major process, contributing 36 %. The remaining 7 % is due the wet and dry deposition, reaction with ozone, and photolysis. The budget is not perfectly closed because the model output frequency (10h) limits the accuracy of the chemical loss calculation, leaving 4 % of sink overestimation within the given simulation year. Lifetimes of carbon suboxide with respect to different chemical

processes are shown in Table 4. It can be seen that reaction with OH is the major sink, followed by cloud droplet uptake and conversion to malonic acid, while reaction with ozone and photolysis play only a minor role.

Simulated mixing ratio distributions are presented in Figure 13 and Figure 14. In general carbon suboxide is below pptv levels, although it can build up to tens of pptv locally. Relatively high mixing ratios occur in Africa, South America, China, and India, where the largest emissions take place. Figure 14 shows the zonal and annual averaged mixing ratios, where the highest values

are close to the surface emissions regions and rapidly declining during transport, mostly due to the fast removal of carbon suboxide by OH. In the southern-equatorial band, carbon suboxide reaches the highest altitudes, in agreement with the strong surface emissions and transport by deep convection. Higher mixing ratios are localized in the southern hemisphere (see Figure 13), due to the major sources being located there. Weak northern sources are situated north of 40°N in Russia, corresponding to the locations of summer boreal forest fires (Kaiser et al., 2012). Carbon suboxide shows intra-annual variability mostly

related to the seasonal cycle of biomass burning.

## 4 Conclusions

In this work, we have studied many aspects of the impact of carbon suboxide, $C_3O_2$, in the atmosphere. The IR and UV spectra of carbon suboxide were measured, showing good agreement with earlier work (Bayes, 1961; Long et al., 1954; Miller and Fateley, 1964; Vander Auwera et al., 1991). The atmospheric photolysis rate constants obtained from the UV spectrum and

assuming a photo-dissociation quantum yield of unity, range from $1 \times 10^{-6}$ to $4.0 \times 10^{-6}$ s$^{-1}$, depending on the altitude, for a zenith angle of 50 degrees.

The reactions of $C_3O_2$ with OH radicals and $O_3$ molecules were studied using relative and absolute methods, as well as theoretically. The rate coefficient for $C_3O_2$ + OH was determined by a relative rate method to be $k_4 = 2.6 \pm 0.5 \times 10^{-12}$ cm$^3$ molecule$^{-1}$ s$^{-1}$ at 295 K, in good agreement with earlier work by Faubel et al. (1977) and supported by theoretical work. The

reaction is predicted to proceed predominantly by OH-addition on the inner carbon atoms, where the final products of this reaction in oxidative conditions are CO and $CO_2$, with the hydroxy radical H-atom converted to an $HO_2$ radical. The rate coefficient for $C_3O_2$ + $O_3$ at 295 K was measured for the first time as $k_6 < 1.5 \times 10^{-21}$ cm$^3$ molecule$^{-1}$ s$^{-1}$, a low value that is supported by theoretical kinetic analysis of the initiation channel. The products of the reaction are predicted to be 2 $CO_2$ + CO, by chemically-activated multi-step decomposition reactions of oxide intermediates.

The Henry's law constant, $K_H$, and the hydration constant, $k_{hyd}$, were measured for the first time. For an acidic aqueous phase, pH < 5, a $K_H$ of $1.08 \pm 0.01$ M atm$^{-1}$ was obtained at 296 K, while for a near-neutral solution, pH = 6-8, a value of $1.56 \pm 0.01$





M atm$^{-1}$ was measured. The hydration rate coefficient ranges from 0.033 s$^{-1}$ for acidic solutions, to 0.043 s$^{-1}$ in near-neutral solution.

The experimental results were incorporated into the EMAC atmospheric chemistry model, simulating the $C_3O_2$ distribution and budget in the atmosphere. It was found that the lifetime of carbon suboxide was determined predominantly by the reaction with OH, accounting for ~60% of its removal. The remainder is nearly exclusively lost to the aqueous phase by conversion to malonic acid and subsequent rainout. The model predicts that malonic acid formed by carbon suboxide hydrolysis remains at sub-pptv levels. This contrasts the interpretation of experimental data by Chebbi and Carlier (1996), who suggest a total source strength leading to tens of pptv of malonic acid being formed in this way. In this work the yield of malonic acid through hydration of carbon suboxide is negligible on the global scale. Wet and dry deposition (as $C_3O_2$), photolysis, and the reaction with $O_3$ have very small contributions. The average lifetime of $C_3O_2$ in the atmosphere is thus found to be around 3 days.

While the current work greatly extends our knowledge on the ambient loss processes of $C_3O_2$, insufficient data is available at this time to determine the tropospheric concentration of carbon suboxide, or its dominant sources, with confidence. The atmospheric model simulations suggest concentrations of the order of pptv, although further sources may yet be discovered. These concentration predictions are compatible with the tentative in-situ measurements.

## 5 Data availability

Additional theoretical calculations for reaction of $C_3O_2$ with OH and $O_3$; the full set of quantum chemical characteristics of all compounds discussed in the theoretical work; the FTIR and UV spectra of $C_3O_2$.

## 6 Competing interests

The authors declare that they have no conflict of interest.

## 7 Acknowledgements

LV was supported by the Max Planck Graduate Center with the Johannes Gutenberg-Universität Mainz (MPGC).

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





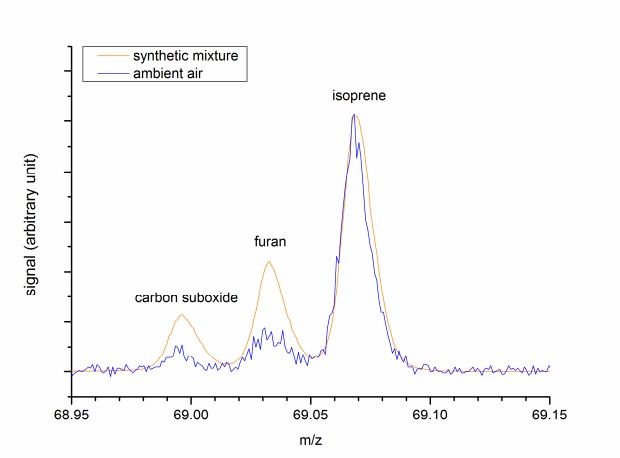

**Figure 1: PTR-TOF-MS mass spectrum of a synthetic mixture of carbon suboxide, furan and isoprene, and of ambient air in Mainz, Germany. The peak heights are normalized to that of isoprene.**

**Figure 2: Beer-Lambert plot of optical density at 264.8 nm versus [$C_3O_2$]. The line is an unweighted proportional fit to the data with a slope of $2.71 \times 10^{-16}$ nm cm$^3$ molecule$^{-1}$.**





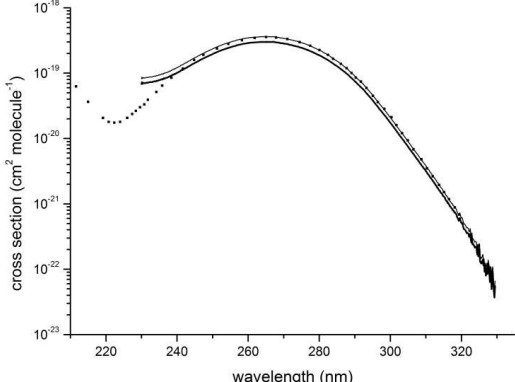

**Figure 3: UV spectrum of $C_3O_2$ in the spectral range 210 to 330 nm. The dots are the spectrum reprinted with permission from Bayes (1961). (Copyright 1961 American Chemical Society). The thick black line is our spectrum; the thin black line was obtained by scaling our spectrum by a factor 1.17 (see text).**

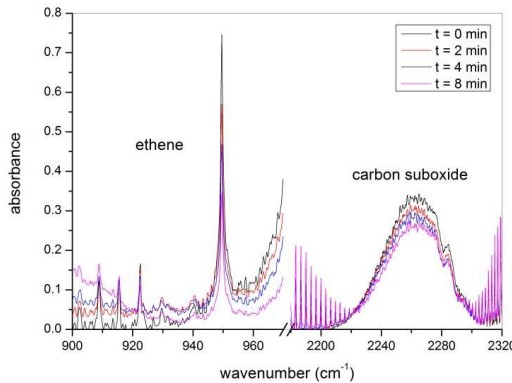

**Figure 4: Depletion of $C_2H_4$ (band centred at ~950 cm$^{-1}$) and $C_3O_2$ (band centred at ~2260 cm$^{-1}$) during the reaction with OH. The black line ($t$=0) is the spectrum obtained before photolysis, the red, blue and purple spectra were obtained after total photolysis**
10 **periods of 2, 6 and 8 minutes, respectively.**



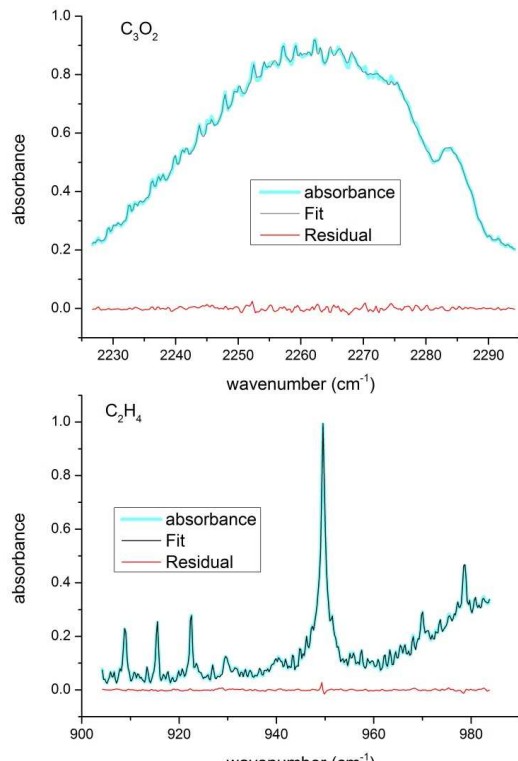

**Figure 5: Examples of least squares fitting to reference spectra (before the reaction was initiated) to obtain the depletion factors for C₃O₂ (upper panel) and C₂H₄ (lower panel) in the relative rate experiments. The red lines show the residuals.**





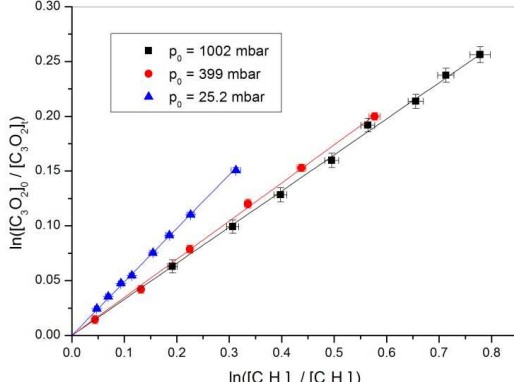

**Figure 6: Depletion factors for C$_3$O$_2$ versus C$_2$H$_4$ during reaction of both traces gases with OH at three different pressures. Error bars are 2$\sigma$ as returned by the least-squares fitting routine. The slope at each pressure yields the relative rate constant ratio k$_4$/k$_5$.**





**Figure 7: Reaction mechanism of the OH-initiated atmospheric oxidation of carbon suboxide**





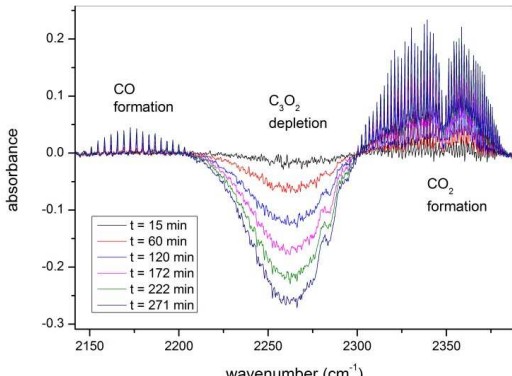

**Figure 8: Depletion of C₃O₂ (band centred at 2260 cm⁻¹) and the formation of CO₂ (2300 to 2375 cm⁻¹) and CO (2150 to 2210 cm⁻¹) at different times during the reaction of C₃O₂ with O₃.**

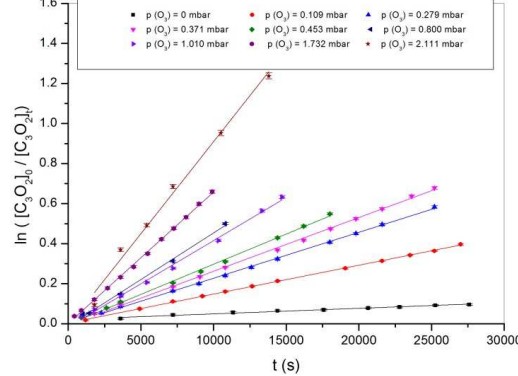

**Figure 9: Relative concentration change of C₃O₂ at various reaction times in the presence of 9 different O₃ concentrations. All experiments were performed at room temperature (~ 295 K) and at a pressure close to 1 bar.**





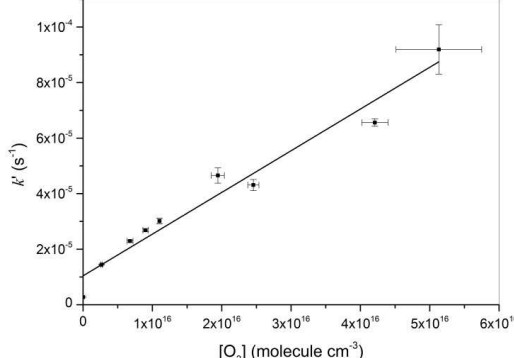

**Figure 10: Pseudo-first-order decay constant of C₃O₂ plotted against the O₃ concentration. The slope is the bimolecular rate constant $k_6$. The horizontal errors bars represent uncertainty in the O₃ concentration based on analysis of three different spectral regions (see text). The vertical error bars are 2σ from the fits to data as shown in Figure 9.**




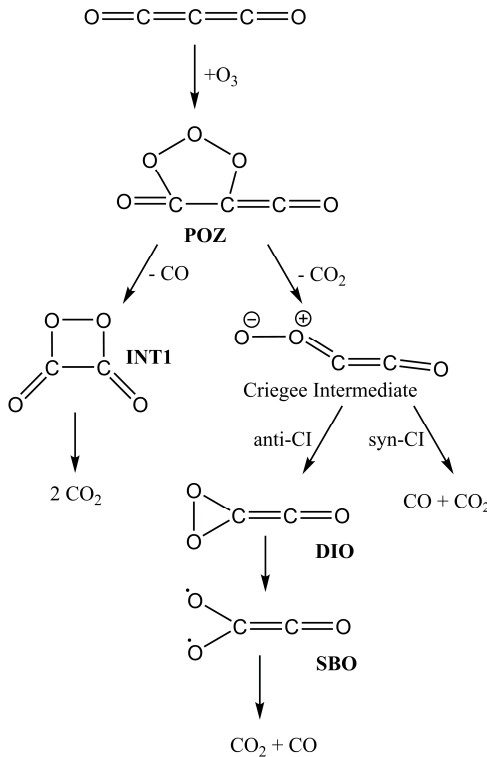

**Figure 11: Simplified mechanism for the reaction of carbon suboxide with O₃. Additional calculations are available in the supporting information.**





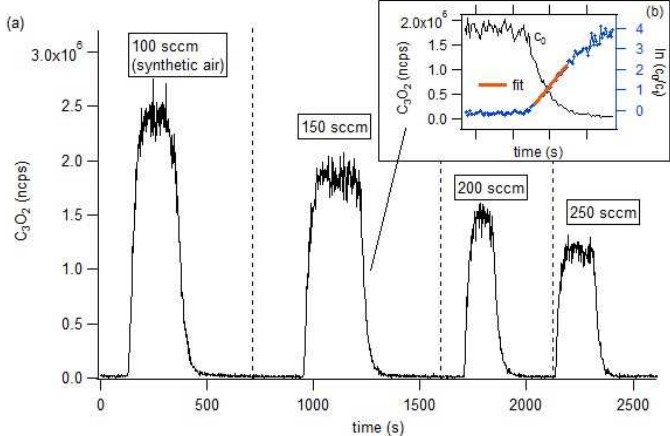

**Figure 12: PTR-TOF-MS measurement of carbon suboxide and different flows of synthetic air through the aqueous solution (pH 6). Inset: example of $C_3O_2$ loss from the solution, the corresponding $\ln(c_0/c_t)$ values, and the linear fit, at 150 cm³ min⁻¹ synthetic air flow.**

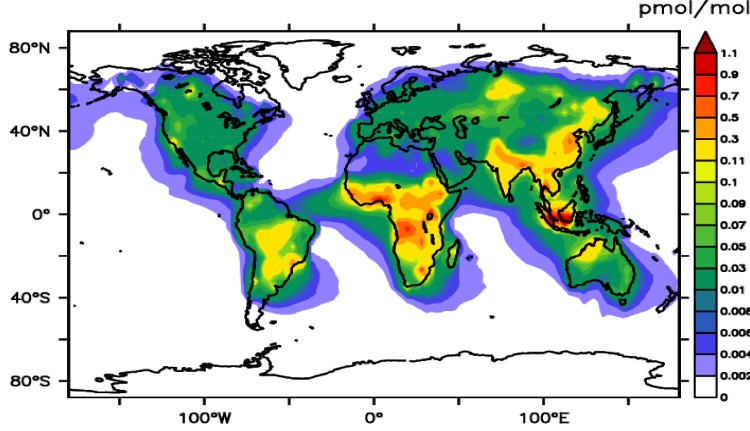

**Figure 13: Model calculated annual near-surface average of carbon suboxide mixing ratios in pmol/mol.**




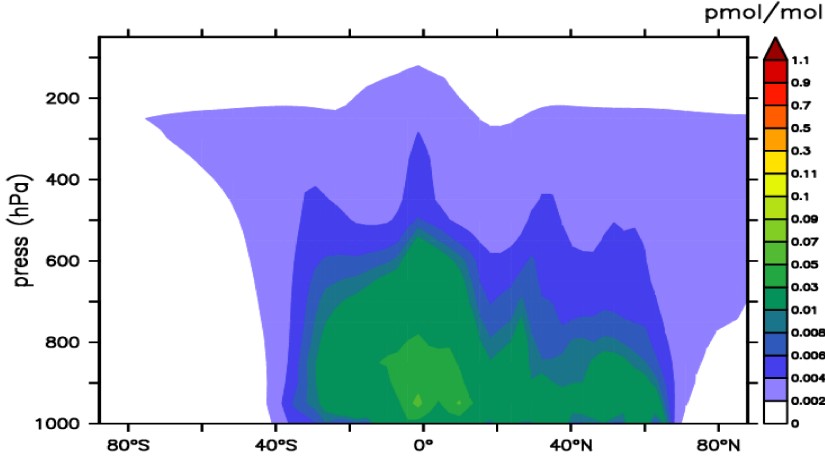

**Figure 14: Model calculated zonal, annual average of carbon suboxide mixing ratios in pmol/mol.**



**Table 1: Experimental values of $k_4 / k_5$, and absolute rate constants ($10^{-12}$ cm$^3$ molecule$^{-1}$ s$^{-1}$).**

| Pressure (mbar) | $k_4 / k_5$ | $k_5$ [b] | $k_4$ |
|---|---|---|---|
| 25.3 | 0.489 ± 0.004 [a] | 4.8 ± 1.0 | 2.3 ± 0.5 |
| 399 | 0.348 ± 0.003 [a] | 7.5 ± 1.5 | 2.6 ± 0.5 |
| 1003 | 0.330 ± 0.002 [a] | 7.9 ± 1.6 | 2.6 ± 0.5 |

[a] Uncertainties are 2 $\sigma$ from weighted, least squares fitting to data such as shown in Figure 6.

[b] Taken from the recommended pressure and temperature dependent value for $k$(OH + C$_2$H$_4$) (Atkinson et al., 2006; Cleary et al., 2006; Fulle et al., 1997; IUPAC Subcommittee on Atmospheric Chemical Kinetic Data Evaluation, 2015; Vakhtin et al., 2003)

**Table 2: Henry's law constants $K_H$ and hydrolysis rate coefficients $k_{hyd}$ obtained as a function of pH, at $T = 296$ K.**

| pH | $K_H$ (M atm$^{-1}$) | $k_{hyd}$ (s$^{-1}$) |
|---|---|---|
| 2 | 1.08 ± 0.01 | 0.033 ± 0.002 |
| 4 | 1.08 ± 0.01 | 0.032 ± 0.004 |
| 6 | 1.56 ± 0.01 | 0.039 ± 0.002 |
| 8 | 1.552 ± 0.003 | 0.043 ± 0.001 |

**Table 3: Global budget of carbon suboxide. The global burden is 0.07 Gg.**

| Sources | Gg/year |
|---|---|
| Biomass burning | +5.48 |
| Biofuel consumption | +2.37 |
| Total source strength | +7.85 |
| **Sinks** | |
| Dry deposition | -0.53 |
| Wet deposition (as C$_3$O$_2$) | -0.00 |
| Reaction with OH | -4.67 |
| Reaction with ozone | -0.01 |
| Photolysis | -0.06 |
| Aqueous phase conversion to malonic acid | -2.89 |
| Total sink strength | -8.14 |

**Table 4: Estimated global average carbon suboxide lifetimes $\tau_{ave}$ from model simulations, for each process separately and for all loss processes combined.**

| Sink | $\tau_{ave}$ |
|---|---|
| Reaction with OH | 5.5 days |
| Aqueous phase conversion | 8.9 days |
| Photolysis | > 1.2 years |
| Reaction with O$_3$ | > 10 years |
| Overall | 3.2 days |