# Peer review of "Atmospheric Chemistry, Sources, and Sinks of Carbon Suboxide, C3O2"

_Atmospheric Chemistry and Physics, 2017_

## Referee Comment (RC1) · P. Seakins (Referee) · 7 Feb 2017

This paper represents a comprehensive study on the fate of C3O2 in the atmosphere. The research has been carried out to a very high standard and is generally clearly presented. There are two areas where I feel some additional input would be useful to enhance the impact of the paper and then there are a number of minor technical points and suggested corrections.

1. Ambient and PTR measurements - The high resolution spectra clearly shows the differentiation between species of protonated mass 69. Is there any possibility that the signal could originate from isotopomers of protonated mass 68, for example 13C or 15N pyrrole? For the ambient measurements in the Mainz region, the only data shown is the spectrum in Fig 1, which I believe was collected over a hour period. Presumably

other data were collected - was C3O2 always present? Was there any evidence of a diurnal cycle or any variability in the C3O2 signal. Perhaps a time series could be included in the supplementary. It would be helpful to state the isoprene concentration in the ambient spectrum component of Fig 1 and the ratio of the peak areas.

2. Given the comprehensive nature of the study, it would be good to provide some more definitive conclusions. Given the relatively short lifetime (n.b. can you check the overall lifetime? I get ~3.4 days not 3.2), the low concentrations and the products formed, are further studies required?

Minor points: Abstract - O3 reaction not measured using relative rate method. Qualify comment on products - the sole carbon products are CO and CO2. HO2 is produced in the OH reaction. p5 line 17. Pathlength was 880 cm on previous page. 880 cm seems to give the value of the cross section reported on p6. Line 30 264.8 nm is presumably lambda max. I don't think this is stated. p6 line 4 multiplier of 10ˆ-19 missing. Line 8 'data' are plural. Line 20 Please include amount of additional NO added, purity etc. Relative rate studies - I was interested in the choice of ethene as a reference compound, the pressure dependence of the reference adds an additional level of complexity to the analysis. Also given the uncertainty in the reference and hence the total rate coefficient, it is not clear that there is any justification in ignoring the low pressure data point. A simple average would give 2.5e-12. Finally the comments on the comparison with Faubel are slightly contradictory - on p7 and 8, the conclusion is that the data are in the same ball park, but not in quantitative agreement and that the relative rate data are likely to be more accurate (I would agree with these conclusions). However, in the conclusions, the data are reported to be in good agreement with Faubel (p14, line 24). p9 line 13 The theoretical predictions of da Silva have been confirmed by experimental studies from our group (Lockhart - JPCA 2013) p10 For the O3 experiments would it be possible to run with an OH radical trap? p13 line 21 - the model input has a rate coefficient of 2.5e-12, not 2.6 References - Need some subscripts in some references. Fig 1 - More details of sampling times, concentrations in the caption. Fig 2 not sure

where the 'nm' comes from Fig 4. Were the fits constrained to go through the origin? The data looks excellent, so don't expect an unconstrained fit would give a significantly different gradient, but should be checked.

---

## Referee Comment (RC2) · Anonymous Referee #2 · 8 Mar 2017

General Comments: This is a thorough study of the atmospheric chemistry of carbon suboxide (C3O2) combining experimental and theoretical results with atmospheric modelling. The experimental results are in qualitative agreement with the previous determination of the rate constant for reaction with OH, and present a new determination of the rate constant for reaction with ozone. The theoretical results roughly support the experimentally determined rate constants and suggest some interesting species for future study. The paper is well written and the results are clearly presented.

Specific Comments: On line 16 of page 4 in the description of the theoretical calculations, it is stated without reference that the energies are expected to be accurate to within ∼2 kcal/mol. Is this true for the barrier heights for ozonolysis as well? Can you include a reference supporting the statement that the M05-2X geometries will be sufficient for future single point calculations? On line 11 of page 7 in the description of the

relative rate study, it is stated that significant reaction with surfaces, CH3ONO, NO, or radicals formed in the dark can be ruled out. Is reaction with HO2 accounted for? Or is it assumed to be slower that R3?

Technical Corrections: Page 8, line 25, "addiction"; SI page 1, at the top of the 2nd paragraph, the acronym "CI" is defined in the text but not in the supplemental information; Figure S2, the caption for the fourth image is somewhat confusing and in the caption for the fifth image "C-C bond is strongly elongates".
* * *

---

## Short Comment (SC1) · 22 Mar 2017

General Comments: The authors may also wish to mention their determination of the Henry's law constant for C3O2 in the abstract (not in the introduction section). The applicability of the dynamic method (Roberts et al., Kames and Schurath), normally employed for stable organic compounds, should be discussed in greater detail. The presented data (Table 2) show a distinct dependence of pH which points to a fast hydrolysis of C3O2. Could the Henry's law constant be dependent on pH or should this dependence be assigned to its hydrolysis? From the inset of Figure 12 it is evident that the depletion phase starts with a fast decrease and turns later to a slower one. This could be interpreted by two different rate constants which are conceivable since the hydrolysis of C3O2 requires two water molecules to produce malonic acid. C3O2 + 2 H2O -> HOOC-CH2-COOH (Eq. 1) The rate constants in Table 2 are calculated

from the initial depletion phase, neglecting the final, smoother descent. This deviation is confirmed by the inset of Figure 12 as well. So the stated measurement errors are not conclusive. Information on the number of runs should be given if a statistical error is meant (see also Figure 12). Are the results obtained from the saturation phase consistent with the depletion phase? It would be helpful to add a new Figure containing the calculation of the rate constants according to Roberts et al. and Kames and Schurath. The source strength of C3O2 should be assessed from two sides: Its formation from natural soils and waters can be neglected due to its fast hydrolysis in these compartments (Huber et al.). A formation in the atmosphere from volatile organic compounds (e.g. phenolic moieties) in the gas phase or on particles is conceivable.

---

## Author Comment (AC1) · 25 Apr 2017

We thank Paul Seakins, Heinfried Schoeler and anonymous referee 2 for the overall positive assessment of the manuscript and the constructive comments. The comments (black), our replies (blue) and the changes made to the manuscript (red) are outlined below.

**Referee 1: Paul Seakins**

1. Ambient and PTR measurements - The high resolution spectra clearly shows the differentiation between species of protonated mass 69. Is there any possibility that the signal could originate from isotopomers of protonated mass 68, for example 13C or 15N pyrrole?

We have added a paragraph (Sect 2.5) to address this point: "In this lower mass range the number of possible and real elemental permutations at a given m/z are limited, making peak assignments highly robust. Protonated 15N pyrrole would appear at a m/z of 69.0465, where it would be expected to interfere with identification and signal allocation of the furan peak, however, is sufficiently removed from the C3O2 peak to allow spectral resolution. The appearance of an organic molecule with a 13C isotopic contribution significant enough to observe on m/z 69 would be accompanied by a large signal on m/z 68, which was not observed during the period over which the measurements of C3O2 were made."

For the ambient measurements in the Mainz region, the only data shown is the spectrum in Fig 1, which I believe was collected over a hour period. Presumably other data were collected - was C3O2 always present? Was there any evidence of a diurnal cycle or any variability in the C3O2 signal. Perhaps a time series could be included in the supplementary. It would be helpful to state the isoprene concentration in the ambient spectrum component of Fig 1 and the ratio of the peak areas.

We now add an additional example spectrum to the supplementary data taken on Cyprus in 2014 extending the mass range out to include pyrrole. This shows graphically that pyrrole does not interfere with this measurement, and the isoprene signal is calibrated for comparison with  $C_3O_2$ . A further dataset taken off the coast of Peru showed no signal for  $C_3O_2$ . This information is now given in the inserted text:

Absence of an interference from pyrrole is further exemplified with an extended mass spectrum using data collected on Cyprus in 2014 with calibrated isoprene and given in the supplementary data section. A third dataset taken in the remote Pacific marine boundary layer showed no  $C_3O_2$  peak.

In supplementary data

**5. Extended Mass Spectrum**

Shown below is an extended mass spectrum taken from data collected in summer 2014 on the island of Cyprus. In this data section isoprene was 35 ppt. Protonated 15N pyrrole would appear at a m/z of 69.0465, where it would be expected to interfere with identification and signal allocation of the furan peak, however, is sufficiently removed from the C3O2 peak to allow spectral resolution. The appearance of an organic molecule with a 13C isotopic contribution significant enough to observe on m/z 69 would be accompanied by a large signal on m/z 68, which was not observed during the period over which the measurements of C3O2 were made in either Mainz or Cyprus. A third campaign dataset was examined from data taken on-board a ship off the coast of Peru but no C3O2 peak was found.

2. Given the comprehensive nature of the study, it would be good to provide some more definitive conclusions. Given the relatively short lifetime (n.b. can you check the overall lifetime? I get \_3.4 days not 3.2), the low concentrations and the products formed, are further studies required?

The difference in the calculated lifetimes is due to the treatment of dry deposition, 3.2 days is the correct value.

We have added a sentence to the conclusions to indicate that more field measurements would be useful to constrain emission rates of  $C_3O_2$ . "While the current work greatly extends our knowledge on the ambient loss processes of  $C_3O_2$ , insufficient data are available at this time to determine the tropospheric concentration of carbon suboxide, or its dominant sources, with confidence. The atmospheric model simulations suggest concentrations of the order of pptv, although further sources may yet be discovered. More measurements of tropospheric  $C_3O_2$  in various locations would help constrain this."

Abstract - O3 reaction not measured using relative rate method.

We have removed the words "using relative rate techniques" in the abstract, which now reads: ".. Rate coefficients for the reactions of C3O2 with OH radicals and ozone were determined as  $k_4 = (2.6 \pm 0.5) \times 10^{-12}$  cm3 molecule-1 s-1 at 295 K (independent of pressure between ~25 and 1000 mbar) and  $k_6 < 1.5 \times 10^{-21}$  cm3 molecule-1 s-1 at 295 K." Qualify comment on products - the sole carbon products are CO and CO2. HO2 is

Qualify comment on products - the sole carbon products are CO and CO2. HO2 is produced in the OH reaction.

We now qualify this (P 8, L10) by stating that "The only (IR-active) stable end-products observed in the present study of  $OH + C_3O_2$  were CO and  $CO_2$ ."

p5 line 17. Pathlength was 880 cm on previous page. 880 cm seems

to give the value of the cross section reported on p6.

880 cm is the correct value (892 was used in a previous version of the absorption cell with internal White optics). We now write: "The UV absorption (230 to 309 nm) at various pressures of  $C_3O_2$  was measured in the optical absorption cell (880 cm path length) described above."

Line 30 264.8 nm is presumably lambda max. I don't think this is stated.

This is correct, we now state that 264.8 nm refers to  $\lambda$ -max. "The dependence of the measured optical density at 264.8 nm ( $\lambda$ max) on the C3O2 concentration is plotted in **Error! Reference source not found.**."

p6 line 4 multiplier of 10-19 missing.

On page 6 we now quote  $3.1 \pm 0.8 \times 10^{-19} \text{ cm}^2 \text{ molecule}^{-1}$

Line 8 'data' are plural.

Singular usage of "data" has been corrected: P6 L9, P11 L8, P15 L11.

Line 20 Please include amount of additional NO added, purity etc.

The sentence "OH radicals were generated by photolysing  $CH_3ONO$  (270-380 nm) in air in the presence of NO" was misleading. We did not add extra NO (to convert HO2 to OH) as sufficient NO is provided from the photolysis of CH3ONO. Not adding extra NO has no consequences (apart from a slightly slower rate of decay of reactants) as HO2 does not react with ethene or C3O2 at a significant rate but will react with itself or with other radicals (e.g. RO2) to make e.g. peroxides. Addition of NO is important in product studies but not essential in relative rate measurements. We now write: OH radicals were generated by photolysing CH3ONO (270-380 nm) in air.

Relative rate studies - I was interested in the choice of ethene as a reference compound, the pressure dependence of the reference adds an additional level of complexity to the analysis. Also given the uncertainty in the reference and hence the total rate coefficient, it is not clear that there is any justification in ignoring the low pressure data point. A simple average would give 2.5e-12.

Ethene is commonly used in OH-relative rate studies as its rate coefficient in 1 bar of air is well known. It also has the advantage of being a small molecule with well resolved lines in the IR and which decomposes to low molecular weight products, which also have simple FTIR spectra. This reduces the risk of products absorbing at the same wavenumber as the reactants, which would bias the analysis.

In the text, we already provide justification for preferring the data at higher pressures (non-linearity in the absorption spectra at low pressure). As the reviewer has already pointed out, an error in the pressure dependent of the OH + C2H4 rate constant may bias the result. This bias will be least at high pressures where the rate coefficient for OH + ethene is arguably best known. Thus, even though the differences between  $2.3 \pm 0.5$  and  $2.6 \pm 0.5$  are statistically insignificant we prefer to quote  $2.6 \times 10^{-12}$  cm3 molecule-1 s-1.

Finally the comments on the comparison with Faubel are slightly contradictory - on p7 and 8, the conclusion is that the data are in the same ball park, but not in quantitative agreement and that the relative rate data are likely to be more accurate (I would agree with these conclusions). However, in the conclusions, the data are reported to be in good agreement with Faubel (p14, line 24).

Agreed. We now write: "The rate coefficient for  $C_3O_2 + OH$  was determined by a relative rate method to be  $k_4 = 2.6 \pm 0.5 \times 10^{-12}$  cm3 molecule-1 s-1 at 295 K, a factor of almost two larger than the value derived by Faubel et al. (1977)."

p9 line 13 The theoretical predictions of da Silva have been confirmed by experimental studies from our group (Lockhart - JPCA 2013)

We have added the reference to the Lockhart et al. paper: "The resulting hydroperoxide acyloxy radical, OCC(OOH)C(O $^{\circ}$ )O, is not an energetic minimum at the chosen level of theory, spontaneously reforming the carboxylic acid peroxy radical; this is contrary to similar reactions in aliphatic carboxylic acid atmospheric oxidation where CO2 is preferentially eliminated (Lockhart et al., 2013; da Silva, 2010). "

p10 For the O3 experiments would it be possible to run with an OH radical trap?

In principal, the addition of an OH scavenger could have reduced the size of this potential bias. The problem is to find a scavenger that does not complicate the IR spectrum and which does not result in peroxy radicals that can also result in re-release of HOx. As we already state, even though it is potentially an upper limit, the rate constant we obtain is so slow that it does not contribute to  $C_3O_2$  loss in the atmosphere and further experiments were considered unwarranted.

p13 line 21 - the model input has a rate coefficient of 2.5e-12, not 2.6

This was a typo. We now write: "The first reaction is with OH, with a rate constant of  $2.6 \times 10^{-12}$  cm3 molecule-1 s-1, the second is photolysis, and the third, minor reaction is with ozone that has a low rate coefficient of  $1.5 \times 10^{-21}$  cm3 molecule-1 s-1."

References - Need some subscripts in some references. Corrected

Fig 1 - More details of sampling times, concentrations in the caption.

The purpose of figure 1 is to show the measurement capability of the instrument in terms of resolution and therefore the uncalibrated ambient and mixed species spectra are simply normalised. Additional data is now provided in the supplementary information

from a summer campaign on Cyprus with a calibrated mixing ratio of isoprene for comparison. This information is included in the figure caption text.

Fig 2 not sure where the 'nm' comes from

The reviewer refers to the caption (page 20). The units should be inverse concentration (cm3 molecule-1). This has been corrected.

Fig 4. Were the fits constrained to go through the origin? The data looks excellent, so don't expect an unconstrained fit would give a significantly different gradient, but should be checked.

Presumably, this comment refers to Figure 6, not Figure 4. The fits were indeed constrained to go through zero. Weighted fitting using x- and y-errors and no constraint resulted in intercepts (-0.002 $\pm$ 0.007 at 1003 mbar, -0.002 $\pm$ 0.003 at 399 mbar and 0.002  $\pm$  0.002 at 25.3 mbar) that were statistically not distinct from zero and slopes (0.334  $\pm$  0.014 at 1002 mbar, 0.353  $\pm$  0.010 at 399 mbar and 0.477  $\pm$  0.015 at 25.2 mbar) that were within 2 % of those listed. We now quote these slopes and intercepts in the paper. These small changes in slope do not affect the final numbers for the OH + C3O2 rate coefficient listed in Table 1. Figure 6 has been modified and the non-zero intercepts are now visible.

**Referee 2**

On line 16 of page 4 in the description of the theoretical calculations, it is stated without reference that the energies are expected to be accurate to within \_2 kcal/mol. Is this true for the barrier heights for ozonolysis as well? Can you include a reference supporting the statement that the M05-2X geometries will be sufficient for future single point calculations?

We have added references to benchmark studies that examined M05-2X and CBS-QB3 (the main methodologies used) to support our values. It should be noted that these benchmarks typically consider a wide variety of reaction classes and heteroatoms, while we are only concerned with H/O/C atoms, which are often better described than less common atoms; as such, the listed accuracy is unlikely to be an underestimate even if it is applied to only a subset of the molecule types used in the benchmarks.

Regarding ozonolysis : it is hard to assess whether the generic accuracy claim is applicable specifically for the C3O2+O3 reaction; I have no knowledge of a benchmark study that specifically studies ozonolysis, though ozonolysis is part of some of the commonly used benchmark databases and could perhaps be lifted out as a subset. One would expect ozonolysis to be less accurate, due to the multi-reference character of the reactant wavefunction. Furthermore, cyclisation reactions specifically have been mentioned as showing a larger than expected spread with CBS-QB3 (Karton, A. and Goerigk, L.: Accurate Reaction Barrier Heights of Pericyclic Reactions: Surprisingly Large Deviations for the CBS-QB3 Composite Method and Their Consequences in DFT Benchmark Studies, J. Comput. Chem., 36(9), 622–632, doi:10.1002/jcc.23837, 2015.) At the same time, these "surprisingly large deviations" reported are still not incompatible with the ~2 kcal mol-1 claim. As the accuracy mentioned in the paper is a general statement, and the ozonolysis would be just a single instance among all reactions examined, we feel that the text is sufficient for its purpose. We now write :

These levels of theory are expected to be accurate within ~2 kcal mol-1 for relative energies (Simmie and Somers, 2015; Somers and Simmie, 2015; Xiu-Juan et al., 2005); this is sufficient to identify the main reaction channels in the atmospheric oxidation of  $C_3O_2$ . If smaller uncertainty intervals are needed in a future study, the geometries listed in the supporting information are expected to be of sufficient quality for more accurate single-point energy calculations (Goerigk and Grimme, 2011; Xu et al., 2011)." On line 11 of page 7 in the description of the relative rate study, it is stated that significant reaction with surfaces, CH3ONO, NO, or radicals formed in the dark can be ruled out. Is reaction with HO2 accounted for? Or is it assumed to be slower that R3? We implicitly (and reasonably) assume that the HO2 radicals present (formed from  $CH_3O + O_2$  following CH3ONO photolysis in air) and which are likely to be present in comparable or higher concentrations than OH do not react with either reactant. We now state this in the manuscript. "In our analysis, we implicitly assume that HO2 radicals (also present) do not contribute significantly to the removal of either reactant as rate coefficient for HO2 reactions with closed shell organics are extremely low at ambient temperatures."

Page 8, line 25, "addiction";

Corrected

SI page 1, at the top of the 2nd paragraph, the acronym "CI" is defined in the text but not in the supplemental information;

Corrected. We now write : "Though we were unable to find formation pathways for these Criegee intermediates (CI), we would like to briefly summarize some additional information on these carbon oxides."

Figure S2, the caption for the fourth image is somewhat confusing and in the caption for the fifth image "C-C bond is strongly elongates".

The captions have been rephrased as follows: Subsequently, at +5.895 Å·amu1/2 past the TS, the oxygen atom in the ozone carbonyl group migrates to the other side of the carbon, while the oxide attacks the central carbon."

"At +7.895 Å·amu1/2 past the TS, the structure starts to form a new cyclic peroxide bond. The endocyclic C–C bond is strongly elongated (1.62 Å), but does not break along the minimum energy path."

Comment: Heinfried Schoeler

The authors may also wish to mention their determination of the Henry's law constant for C3O2 in the abstract (not in the introduction section).

Agreed. We now write: "The UV absorption spectrum and the interaction of  $C_3O_2$  with water (Henry's law solubility and hydrolysis rate constant) were also investigated, enabling it photodissociation lifetime and hydrolysis rates, respectively, to be assessed."

The applicability of the dynamic method (Roberts et al., Kames and Schurath), normally employed for stable organic compounds, should be discussed in greater detail. The presented data (Table 2) show a distinct dependence of pH which points to a fast hydrolysis of C3O2. Could the Henry's law constant be dependent on pH or should this dependence be assigned to its hydrolysis? From the inset of Figure 12 it is evident that the depletion phase starts with a fast decrease and turns later to a slower one. This could be interpreted by two different rate constants which are conceivable since the hydrolysis of C3O2 requires two water molecules to produce malonic acid. C3O2 + 2 H2O -> HOOC-CH2-COOH (Eq. 1) The rate constants in Table 2 are calculated from the initial depletion phase, neglecting the final, smoother descent. This deviation is confirmed by the inset of Figure 12 as well. So the stated measurement errors are not conclusive. Information on the number of runs should be given if a statistical error is meant (see also Figure 12). Are the results obtained from the saturation phase consistent with the depletion phase? It would be helpful to add a new Figure containing the calculation of the rate constants according to Roberts et al. and Kames and Schurath.

Equation 4 is used to describe the first order decay of  $C_3O_2$  in the liquid as a result of purging and hydrolysis. The decay is expected to be exponential in the bubble column as it is representative of the combined effect of both purging and hydrolysis. The inset in figure 12 displays the expected exponential decay after removal of the  $C_3O_2$  as well

as a semilog plot of  $\ln(C_0/C_t)$  which is linear (red fit). Equation 4 allows both the effective Henry's constant and the first-order rate loss of hydrolysis to be determined 'independently'. Therefore, our results suggest that both the Henry's law constant and the first-order rate loss of hydrolysis are dependent on pH. The first-order rate loss of hydrolysis is the effective rate loss describing the individual hydrolysis processes that are occurring, such as the various hydrolysis steps the reviewer mentions. Thus first-order rate loss of hydrolysis could be used to determine the individual hydrolysis rate constants, see Borduas et al. 2016, however, that is beyond the scope of this work as liquid phase concentration of the hydrolysis products would need to be known. Lastly, the calculation of the rate constants does not deviate from the method used in the work cited and for that reason we have chosen to not include a detailed accounting of the calculation.

Borduas, N., Place, B., Wentworth, G. R., Abbatt, J. P. D., and Murphy, J. G.: Solubility and reactivity of HNCO in water: insights into HNCO's fate in the atmosphere, Atmos. Chem. Phys., 16, 703-714, 10.5194/acp-16-703-2016, 2016

The source strength of C3O2 should be assessed from two sides: Its formation from natural soils and waters can be neglected due to its fast hydrolysis in these compartments (Huber et al.). A formation in the atmosphere from volatile organic compounds (e.g. phenolic moieties) in the gas phase or on particles is conceivable.

We add the omitted reference (Huber et al. 2007), mention the likely negligible surface soil-to-air flux, and note the potential of airborne production via aromatic species and particle surface reactions.

Production of  $C_3O_2$  has been observed in laboratory studies of soil (Huber et al. 2007), however, in-situ hydrolysis likely precludes significant surface soil-to-air fluxes. At this time, we do not propose atmospheric  $C_3O_2$  sources although these may occur via decomposition of aromatic entities or particle surface reactions, but instead adhere to the single source category described in the literature (biomass burning), discussed in more detail below

---

## Author Comment (AC2) · 25 Apr 2017

The comment was uploaded in the form of a supplement:
http://www.atmos-chem-phys-discuss.net/acp-2017-49/acp-2017-49-AC2-
supplement.pdf

---

## Author Comment (AC3) · 25 Apr 2017

The comment was uploaded in the form of a supplement: http://www.atmos-chem-phys-discuss.net/acp-2017-49/acp-2017-49-AC3-supplement.pdf

---

## Author Response (AR2)

We thank the editor for spotting the typos and formatting errors (now corrected) listed below.

P1, L11: is largely unknown -> are largely unknown

Corrected

P1, L13-14: use kOH instead of k4 and kO3 instead of k6 in the abstract

Corrected

P1, L20: e.g. by biomass burning -> for example, in presence of biomass burning

Corrected to e.g. influenced by biomass burning

P2, L3: (Huber et al., 2007), is emitted -> (Huber et al., 2007). It is emitted

Corrected

P2, L10: of carbon suboxide -> of carbon suboxide in more detail

Corrected

P2, L19: in a similar manner to described previously -> as described previously

Not corrected, we do not follow the exact method described previously as would be implied by "as described previously", the difference is noted in the next sentence.

P3, L30: search for all instances of m/z and italicize the ones that are not italicized already

Corrected

P3, L31: mass resolution -> mass resolving power

Corrected

P4, L4: e.g. 1 hour -> e.g., 1 hour (also search for other instances of "e.g." which should be surrounded by commas)

This is not necessary since Oxford English grammar is accepted by ACP. (https://en.oxforddictionaries.com/definition/e.g.). Note this seems to be stylistic preference (rather than a rule of grammar) found in American English and not in British, which may explain the difference in opinion.

P5, L8: i.e. -> i.e., [also search for other instances of "i.e." which should be surrounded by commas]

This is not necessary since Oxford English grammar is accepted by ACP. (https://en.oxforddictionaries.com/definition/i.e.

Note this seems to be stylistic preference (rather than a rule of grammar) found in American English and not in British, which may explain the difference in opinion.

P5, L12: covers -> covered [there are a few other issues with consistency of verb tenses in this section]

Corrected

P6, L9: Boyles law -> ideal gas law [Boyles law is something else]

Corrected

P9, L23: Is the statement that INT8 has a fairly mobile hydrogen atom a guess or based on calculations? It is not clear from the text.

This comes from calculations. We now state that the H-shift endothermicity was calculated, as the low isomerisation energy is the main factor driving the mobility.

P10, L12: concentrations were -> partial pressure was

Corrected

P11, L20: There is no previous experimental data -> There are no previous experimental data

Corrected

P12, L29: acid -> acidic

Corrected

P15: Figures 14 appears here twice and also appears again on page 32. Please fix.

This is very odd, a software bug in our pdf maker program, it has been corrected of course.

All formulas: did not display correctly in the PDF but hopefully ACP formatting will fix that

We will pay attention to this in the proofs but in the most recent pdf they are all correct.

All figures some of the figures appear to have low resolution – generating images with higher dpi is recommended for the final publication

All figures will be sent separately at least 300 dpi.

Figure 1: figure background color should be white

In the submitted version it is white